# A ROCK Inhibitor Promotes Graft Survival during Transplantation of iPS-Cell-Derived Retinal Cells

**DOI:** 10.3390/ijms22063237

**Published:** 2021-03-22

**Authors:** Masaaki Ishida, Sunao Sugita, Kenichi Makabe, Shota Fujii, Yoko Futatsugi, Hiroyuki Kamao, Suguru Yamasaki, Noriko Sakai, Akiko Maeda, Michiko Mandai, Masayo Takahashi

**Affiliations:** 1RIKEN Center for Biosystems Dynamics Research, Laboratory for Retinal Regeneration, Kobe 650-0047, Japan; masaaki.ishida@riken.jp (M.I.); makabe.k1@gmail.com (K.M.); shota.fujii@gmail.com (S.F.); yoko.futatsugi@riken.jp (Y.F.); hiroyuki_retina_galileogalilei@yahoo.co.jp (H.K.); suguru-yamasaki@ds-pharma.co.jp (S.Y.); noriko.sakai@riken.jp (N.S.); akiko.maeda@riken.jp (A.M.); michiko.mandai@riken.jp (M.M.); retinalab@ml.riken.jp (M.T.); 2Graduate School of Medicine, Kyoto University, Kyoto 606-8501, Japan; 3Vison Care Inc, Kobe 650-0047, Japan; 4Department of Ophthalmology, Kawasaki Medical School, Okayama 701-0114, Japan; 5Regenerative and Cellular Medicine Kobe Center, Sumitomo Dainippon Pharma Co., Ltd., Kobe 650-0047, Japan

**Keywords:** ROCK inhibitor, iPS cells, retinal pigment epithelium, transplantation

## Abstract

Currently, retinal pigment epithelium (RPE) transplantation includes sheet and single-cell transplantation, the latter of which includes cell death and may be highly immunogenic, and there are some issues to be improved in single-cell transplantation. Y-27632 is an inhibitor of Rho-associated protein kinase (ROCK), the downstream kinase of Rho. We herein investigated the effect of Y-27632 in vitro on retinal pigment epithelium derived from induced pluripotent stem cells (iPS-RPE cells), and also its effects in vivo on the transplantation of iPS-RPE cell suspensions. As a result, the addition of Y-27632 in vitro showed suppression of apoptosis, promotion of cell adhesion, and higher proliferation and pigmentation of iPS-RPE cells. Y-27632 also increased the viability of the transplant without showing obvious retinal toxicity in human iPS-RPE transplantation into monkey subretinal space in vivo. Therefore, it is possible that ROCK inhibitors can improve the engraftment of iPS-RPE cell suspensions after transplantation.

## 1. Introduction

Recently, clinical trials using human embryonic stem cell/induced pluripotent stem cell (ESC/iPSC)-derived tissues have shown promise [1,2]. For cultures of human ESC/iPSC, inhibitors of Rho-associated protein kinase (ROCK) have been used. ROCK is a downstream kinase of the small GTPase Rho, and its inhibition suppresses apoptosis of human ESC/iPSC [3]. It also facilitates proliferation and extended passages of human ESC/iPSC-derived retinal pigment epithelium (ESC-RPE/iPS-RPE) while inhibiting epithelial-to-mesenchymal transition through various pathways [4,5]. ROCK inhibitors are also used as effective drugs for some diseases. For example, in ophthalmology, eye drops for glaucoma such as ripasudil have been approved in Japan [6].

Okumura et al. demonstrated that the ROCK inhibitor Y-27632 promotes adhesion and proliferation of cynomolgus monkey-derived corneal endothelial cells [7]. Subsequently, Kinoshita et al. demonstrated that in a patient with corneal disorders, the injection of human donor-derived corneal endothelial cells together with Y-27632 was effective for corneal repair [8]. In retinal studies, Zheng et al. (2004) studied the effect of Y-27632 on transplantation of primary RPE cells in rabbit models [9]. Moreover, to promote RPE differentiation from human iPSC, Ye et al. used and added 10 μM Y-27632 to culture medium for 18 days [10]. Despite these lines of evidence showing the efficacy of Y-27632 in clinical transplantation, there are no studies so far regarding the usage of Y-27632 for iPS/ES-RPE cell transplantation. Therefore, in this study, we investigated the effects of the ROCK inhibitor Y-27632 on iPS-RPE culture and on transplantation of iPS-RPE cell suspensions into the subretinal space. In particular, the evaluation of retinal toxicity and the change in the engraftment amount of Y-27632-pretreated iPS-RPE cells were confirmed by subretinal administration of this ROCK inhibitor.

First, we examined the effects of Y-27632 in vitro, such as the inhibition of apoptosis, improvement of adhesion, and promotion of pigmentation of iPS-RPE cells. We also examined whether the immunogenicity of Y-27632-treated iPS-RPE cells was altered or whether Y-27632 itself had immunosuppressive capacity. We also checked the toxicity of Y-27632 on retinal tissues in vivo with a focal electroretinogram and immunohistochemistry (IHC). Then we transplanted allogenic iPS-RPE cell suspensions with immunosuppressants into the subretinal space of monkeys with or without Y-27632, and showed that the administration of Y-27632 during transplantation enhanced the engraftment of the transplanted cells.

## 2. Results

### 2.1. Promotion of Cell Proliferation and Cell Adhesion in Human iPSC-RPE Cells with Y-27632

We first examined the proliferation and cell adhesion in iPSC-derived RPE cells in the presence of Y-27632. As revealed in Figure 1A, iPS-RPE cells in the presence of Y-27632 clearly showed polygonal and hexagonal proliferation, whereas cells without Y-27632 showed poor proliferation in comparison. In a FACS analysis of cell proliferation (Ki-67 positive cells), Y-27632-treated iPS-RPE cell numbers were significantly greater than those of nontreated cells (Figure 1B). In addition, we also examined the cell adhesion of Y-27632-treated iPS-RPE cells. Compared with nontreated RPE cells, we found many PKH-positive Y-27632-treated RPE cells on the plate by microscopic observation (Figure 1C). Similarly, the adhesion of Y-27632-treated RPE cells was significantly higher than that of nontreated cells by PKH-positive cell determination (Figure 1D). These results indicated the promotion of proliferation and cell-to-cell adhesion in iPS-RPE cells when they were pretreated with Y-27632 in vitro.

### 2.2. Inhibition of Apoptosis, Promotion of Pigmentation, and Detection of ZO-1, Phospho-Myosin Light Chain 2 in Y-27632-Treated iPS-RPE Cells

As a next step, we investigated whether Y-27632-treated iPS-RPE cells showed inhibited cell death, e.g., apoptosis. For the assay, iPS-RPE cells were treated with Y-27632 (10 µM) for 24 h. Compared with nontreated iPS-RPE cells (253G1, Ff-I01 lines), 10 µM Y-27632-treated RPE cells exhibited fewer annexin V-positive cells via flow cytometry (Figure 2A). As revealed in Figure 2B, statistical analysis showed that there were significant differences between Y-27632-treated and nontreated RPE cells via annexin V. We also examined the difference of the pigmentation in Y-27632-treated iPS-RPE cells. Compared with nontreated cells, RPE cells in the presence of Y-27632 had much pigmentation (Figure 2C). In addition, Y-27632-treated iPS-RPE cells clearly expressed ZO-1 (a tight junction marker for RPE cells) compared with nontreated cells (Figure 2D).

We next examined whether the ROCK activity was regulated in Y-27632-pretreated RPE cells by IHC. We observed the high expression of phospho-myosin light chain 2 (pMLCII) in iPS-RPE cells. However, the signal diminished in Y-27632-pretreated iPS-RPE cells (Figure 2E). Appendix A shows that the reduction of phosphorylation started 15 min after the addition of Y-27632 in RPE cells and continued for at least 24 h. In a dose-response analysis (Appendix A), after adding 100 µM of Y-27632, there was evidence that the RPE cells were damaged (i.e., stained with phalloidin), while a concentration of 10 µM seemed optimal. According to these results, we chose a Y-27632 concentration of 10 µM for subsequent in vitro and in vivo assays.

### 2.3. Suppression of Production of Inflammatory Cytokines/Chemokines in Y-27632-Treated iPS-RPE Cells

Next, we examined the production of inflammatory cytokines/chemokines by iPS-RPE cells in the presence of Y-27632. We checked IL-6, TNF-α, CXCL11/I-TAC, and CCL2/MCP-1 in Y-27632-treated iPS-RPE cells using qRT-PCR and ELISA. In the qRT-PCR analysis, Y-27632-treated iPS-RPE cells had reduced mRNA expression for *IL-6*, *CXCL11/I-TAC*, and *CCL2/MCP-1*, although the cells did not express *TNF-α* mRNA (Appendix A). In a dose-dependent assay, Y-27632-treated iPS-RPE cells produced less *CCL2/MCP-1* mRNA (Appendix A) and protein (Appendix A) compared with nontreated RPE cells. These results indicated that Y-27632-treated RPE cells produced fewer inflammatory cytokines/chemokines.

### 2.4. Low Immunogenicity in Y-27632-Treated iPS-RPE Cells

We next examined the expressions of HLA-class II molecules (HLA-DR, DQ, and OP) in Y-27632-treated iPS-RPE cells. Before the assay, the RPE cells were pretreated with recombinant IFN-γ for 48 h. Compared with cells without Y-27632, the expressions of HLA-class II in Y-27632-treated iPS-RPE cells were poor according to FACS analysis (Figure 3A). Moreover, there were significant differences between Y-27632-treated vs. nontreated RPE cells in mean fluorescence intensity for the expression of HLA-class II molecules (Figure 3A).

We also performed the lymphocytes–grafts immune reaction (LGIR) test using allogeneic lymphocytes and Y-27632-treated iPS-RPE cells. Details of the LGIR assay, including schematic presentation, are shown in Figure 3B. For nontreated RPE cells in the LGIR test (Figure 3C), there was proliferation of CD4^+^/Ki-67^+^ (proliferative helper T cells) in peripheral blood mononuclear cells (PBMC), and there was less proliferation of CD4^+^/Ki-67^+^ in PBMC with Y-27632-treated RPE cells. In addition, there was a statistically significant difference between nontreated and Y-27632-treated iPS-RPE cells. Moreover, as shown in Figure 3C, there was less proliferation of CD8^+^/Ki-67^+^ (proliferative cytotoxic T cells) with Y-27632-treated RPE cells compared with the PBMC plus RPE cells (no Y-27632). Taken together, these results indicated that Y-27632-treated iPS-RPE cells had low immunogenicity, at least in vitro.

Next, we examined whether Y-27632 directly suppressed the activation of lymphocytes in vitro. As a result, Y-27632 directly suppressed cell proliferation of CD4^+^/Ki-67^+^ (proliferative helper T cells), CD8^+^/Ki-67^+^ (proliferative cytotoxic T cells), and CD11b^+^/Ki-67^+^ (proliferative monocytes) when mixed lymphocyte reactions (MLR) were performed (Appendix A). In addition, Y-27632 significantly suppressed the production of IFN-γ inflammatory cytokines (Appendix A), suggesting that Y-27632 has the ability to inhibit inflammation.

### 2.5. Toxicity of In Vitro iPS-RPE Cells and In Vivo Retinas in a Cynomolgus Monkey by Y-27632 Treatment

In order to test the quality of Y-27632-treated iPS-RPE cells and Y-27632 toxicity, we performed several quality control tests. First, we checked phagocytosis by Y-27632-treated iPS-RPE cells, because phagocytosis is one of the most important functions of RPE cells. The iPS-RPE cells were cultured with shed photoreceptor rod outer segments (ROS) for 24 h and analyzed by flow cytometry. As a result (Figure 4A), there were no significant differences between Y-27632-treated and nontreated RPE cells in mean fluorescence intensity for the expression of ROS, indicating that phagocytosis did not function differently between two RPE cell groups (Figure 4A).

Second, we examined the expression of RPE-specific markers VEGF-A, PEDF, RPE65, bestropin-1, Pax6, and tyrosinase in Y-27632-treated iPS-RPE cells. The qRT-PCR showed no statistically significant differences between Y-27632-treated and Y-27632-nontreated RPE cells, as shown in Appendix A. In IHC, both Y-27632-treated and nontreated RPE cells clearly expressed VEGF (Appendix A) and PEDF (Appendix A) on their surfaces, and there were no significant differences between Y-27632-treated and nontreated cells in mean fluorescence intensity for the expression of VEGF (Appendix A). On the other hand, Y-27632-treated RPE cells significantly expressed PEDF according to IHC compared with RPE cells without Y-27632 (Appendix A).

As a next step, we evaluated the toxicity to retinal tissues after Y-27632 intraocular administration. For this assay, we used the eyes of normal adult cynomolgus monkeys. We administrated Y-27632 in saline into the left eye, and saline only as a control into the right eye, and evaluated the retinas including the RPE layers using focal ERG (retinal sensitivity), color fundus, fundus autofluorescence (AF), fluorescence fundus angiography (FA), optical coherence tomography (OCT), and IHC in retinal sections. First, we measured focal ERG (fERG) in the retinas of both eyes, Y-27632 in the left and saline only in the right eye. Compared with preoperative data, fERG was temporary decreased in both retinas one month after administration, and then it recovered to the baseline 3–4 months postinjection (Appendix A). There were no significant differences between them after a 6-month evaluation, indicating that Y-27632 administration in the eye did not change retinal sensitivity. Similarly, there were no differences between Y-27632 and control retinas in color fundus (Appendix A), AF, FA, and OCT (data not shown). Similar to control retinas, retinal sections from those administered with Y-27632 showed no damage according to hematoxylin and eosin (H&E) evaluation (Figure 4B). In IHC (Appendix A), neural retinas clearly expressed RBP3/IRBP (a marker for neural retina and RPE cells), and the RPE cell layer clearly expressed collagen IV (a marker for basement membrane). These results indicated that there was no damage and no toxicity to retinas after Y-27632 administration. 

### 2.6. Transplantation of Allogeneic iPS-RPE Cells Together with Y-27632 in an In Vivo Animal Model (a Cynomolgus Monkey)

In the present study, we used an in vivo monkey transplantation model to assess the efficacy of Y-27632. We first transplanted allogeneic iPS-RPE cells with Y-27632 into cynomolgus monkey eyes from MHC haplotype-mismatched donors with immunosuppression. Before the transplantation, we administrated oral cyclosporin A. As shown in Figure 5, we found pigmented graft cells in the retinas of the monkey left eyes. Also, there were no rejection signs according to FA and OCT at 4 and 12 weeks (Figure 5). There were no inflammatory signs in the retinas, and grafted cells survived and expanded over 6 months of observation. In contrast, the control eyes (right eyes) in the same monkeys were quite different. Without Y-27632, we transplanted allogeneic iPS-RPE cells, the same 46a RPE cell line, with the same method. In examination of the color fundus, we found pigmented graft cells, but they were not expanded (like a clump) in the retinas of monkey right eyes at 4 weeks (Appendix A, left panel) and 24 weeks (Appendix A, middle panel). In addition, there were no rejection signs according to FA (Appendix A, right panel) and OCT (Appendix A). Compared with retinal sections in the right eye without Y-27632 (Appendix A), the in vivo expanded graft RPE cells were seen throughout the retinal sections from the left eye (Appendix A).

After sacrifice, we performed H&E staining using retinal sections from transplanted monkeys. In the results of staining in both eyes, there was graft survival of iPS-RPE cells in the subretinal space (Figure 6A,B). Importantly, the expanded RPE cells in the left eyes (with Y-27632 RPE cell transplantation) took on a sheetlike appearance (Figure 6B). Using confocal microscopy, we found PKH-labeled iPS-RPE cells (Figure 6C). Compared with retinal sections in the right eyes without Y-27632 (Appendix A), the expanded RPE cells were seen through the retinal sections from the left eyes with Y-27632 (Appendix A). There were no signs of immune attack on the retina, including no invasion of CD3^+^ T cells nor MHC-II^+^ antigen-presenting cells, and there were few Iba1^+^ microglia/macrophages in the choroid throughout the sections, whereas there was graft survival of transplanted RPE cells (Figure 6C1–C3). We obtained similar results showing poor invasion of other inflammatory factors/cells such as IgG, NKG2A (NK cells), and CD20 (B cells) (data not shown).

We also measured the RPE cell graft area in transplantations with or without Y-27632. After transplantation, AF images showed a dark area that corresponds to the grafted RPE cells, especially in the left eyes (Figure 6D). Compared with transplantations without Y-27632, the cell graft area was 5-fold greater in transplantations with Y-27632 (Figure 6D), i.e., 0.846 mm^2^ without Y-27632 vs 4.594 mm^2^ with Y-27632. These results indicated that Y-27632 RPE cell transplantation may help to prolong graft survival.

### 2.7. Transplantation of Allogeneic iPS-RPE Cells with Y-27632 in MHC Haplotype-Matched or Steroid-Administrated MHC Haplotype-Mismatched Cynomolgus Monkeys

To confirm the in vivo efficacy of Y-27632, we next transplanted allogeneic iPS-RPE cells in the presence of Y-27632 into cynomolgus monkey eyes in an MHC haplotype-matched donor or in an MHC haplotype-mismatched donor with immunosuppressive drugs (steroid). As revealed in Figure 7A–E, there were pigmented graft cells in the monkey left eye in the MHC-matched donor. The grafted cells survived and there were no rejection signs in OCT at 8 weeks (Figure 7C). After sacrifice, we performed H&E staining using retinal sections from transplanted. In addition, H&E staining showed graft survival of iPS-RPE cells in the subretinal space (Figure 7D). In confocal microscopic images, we found PKH-labeled iPS-RPE cells throughout the retinal sections (Figure 7E). We obtained similar results with an MHC-mismatched donor with immunosuppressive drugs and local steroid administration (Figure 7F–J). Examinations such as color fundus, AF, OCT, H&E staining, and IHC staining indicated graft survival of iPS-RPE cells in the subretinal space.

## 3. Discussion

ROCK inhibition promotes cell adhesion and proliferation, and inhibits apoptosis in some cells/tissues in vitro. The first trial using a ROCK inhibitor for transplantation therapy of cornea endothelium dysfunction was made by Koizumi et al. [11], in which transplantations of cornea endothelial cells into the anterior chamber under the presence of a ROCK inhibitor showed good results in an in vivo animal model. Then Koyanagi et al. reported that a ROCK inhibitor prevented apoptosis of ESC-derived neural precursor cells in vitro and also suppressed acute apoptosis of grafted cells after transplantation in an animal model [12]. Recently, clinical application of a ROCK inhibitor in patients with corneal diseases has been reported [8], and it resulted in increased corneal endothelial cell density in patients with bullous keratopathy after cell injection. Since transplantation using a ROCK inhibitor has been successful in clinical trials, in the present study we investigated the effect of Y-27632 on our established iPS-RPE culture and iPS-RPE cell transplantation system.

The activation of ROCK contributes to survival in several cell types, and it blocks apoptosis in embryonic stem cells, cardiomyocytes, and corneal cells [3,13,14,15,16]. Croze et al. reported that the inhibition of ROCK allows for extended passages of ES/iPS-RPE with appropriate morphology and expression of key RPE markers without an increase in pluripotent, fibroblastic, or endothelial transcripts [4]. In addition, although fetal RPE is different from ESC/iPS-RPE in origin, similar results in this cell type were already reported [17]. Indeed, in our study, Y-27632 promoted the adhesion of iPS-RPE cells and inhibited their apoptosis in vitro. We also found an improvement in the pigmentation and morphological characteristics of iPS-RPE cells. The expressions of RPE-specific markers such as VEGF-A, PEDF, RPE65, bestropin-1, Pax6, and tyrosinase showed no significant differences with or without Y-27632. These results suggest that Y-27632 may provide faster and more efficient protocols to produce high-quality iPS-RPE cells required for clinical trials.

There are several reports of oral or eye-drop administration of ripasudil (K-115) in mouse models that have shown neuronal protection of RGC [18] and treatment efficacy in a retinal hypoxic neovascular disease model [19]. However, there are no studies so far on the subretinal administration of Y-27632. Therefore, we injected Y-27632 into monkey subretinal spaces and confirmed that there was no morphological or electrophysiological damage in the retinal tissues compared to control injections. Although vitrectomy for the injection caused small damage to the retinal tissues, and indeed a marked decline of the focal ERG responses was shown after the operations, they recovered under both conditions, which is consistent with a previous report [20]. In conclusion, subretinal administration of Y-27632 did not cause any damage and no toxicity to the retinal tissues.

Koyanagi et al. reported that Y-27632 inhibited apoptosis of grafted neural precursor cells in mouse brains [12]. However, there are no studies regarding the effect of Y-27632 on the retina and on apoptosis of grafted retinal cells in vivo. Therefore, in our study, we researched the efficacy and safety of Y-27632 upon transplantation of iPS-RPE cell suspensions in vivo. We studied immune rejection and graft survival of iPS-RPE cell transplants by optical coherence tomography, fundus autofluorescence, fluorescence angiography, and IHC. In fundus autofluorescence, grafted RPE cells with dark pigmentation correspond to low fluorescent areas. Fundus autofluorescence measurements showed that the administration of Y-27632 gave larger grafted areas. Using IHC, we observed sheeted RPE grafts under the administration of Y-27632, which was not observed without Y-27632.

ROCK inhibition has immunosuppressive effects. For example, previous reports showed that a ROCK inhibitor suppressed allogeneic immune responses [21] and graft versus host disease (GVHD) [22,23]. Tharaux et al. examined alloantigen-induced proliferation in MLR assays with a ROCK inhibitor. We also had similar results with our MLR in the presence of Y-27632 (Appendix A). Moreover, in our lab, we established the lymphocytes–grafts immune reaction (LGIR) test to measure immune rejection in vitro by using host immunocompetent cells (mainly lymphocytes) and transplant cells (target iPS-RPE cells) [24,25,26,27,28]. With this system, we have demonstrated that T cells proliferate when MHCs are mismatched between the donor and recipient. In the present study, we showed that T cells proliferated when Y-27632 was not used. However, when Y-27632 was administrated, T cells did not proliferate even though MHC was mismatched. As shown in our in vitro assays, MHC class II on RPE cells was downregulated by Y-27632, suggesting that immune rejection of a graft would be reduced by Y-27632. Since RPE cells exhibit immunogenic properties [29], we have to consider how to reduce intraocular inflammation after transplantation. To test the effect of Y-27632 against immune rejection, further in vivo experiments will be required to see whether administration of Y-27632 during transplantation allows for fewer or no immunosuppressants.

## 4. Experimental Procedures

### 4.1. RPE Cell Culture and Reagents

For differentiation into RPE cells, human iPS cells (TLHD2, Ff-I01, 453F2, 454E2, QHJIs01-01, and 253G1 lines) were cultured as described in previous reports [27]. After establishment of iPS-RPE cells, the medium was changed to Dulbecco’s modified Eagle’s medium (DMEM) supplemented with B27 supplement (Invitrogen) and 2 mM L-glutamine (Sigma-Aldrich, St. Louis, MO, USA), and the RPE colonies were transferred to CELLstart^TM^ (Invitrogen, Carlsbad, CA, USA)-coated plates or dishes in B27 medium supplemented with basic FGF (Wako, Osaka, Japan) and SB431542 (Sigma-Aldrich) and cultured until confluence [27]. 

The ROCK inhibitor Y-27632 (1, 10, 50, or 100 µM: Wako) was added to the DMEM medium plus 10% fetal bovine serum (FBS) in some assays.

We also prepared two iPSC from normal cynomolgus monkeys (*Macaca fascicularis*), 46a iPSC from a Cyn46 MHC heterozygous monkey, and 1121A1 iPSC from a HT-1 MHC homozygous monkey [25,30]. We transplanted two iPS-RPE cells (46a and 1121A1) with Y-27632 to monkey models.

### 4.2. Immunohistochemistry and Microscopy

For evaluation of pigmentation in RPE cells, iPS-RPE cells (454E2) were cultured for 4 weeks with or without Y-27632 10 µM. We added Y-27632 once per week in the 4-week cultures, and evaluated with the pictures by microscopy.

For evaluation of cell adhesion in RPE cells, RPE cells differentiated from iPS cells (253G1) were labeled with PKH fluorescent dyes (567 nm: Sigma-Aldrich: catalog no. PKH26GL) and seeded on the 24-well plastic plates without coating materials (1.0 × 10^6^ cells/well) in the presence or absence of 10 μM Y-27632. These cells were incubated at 37 °C for the indicated time periods, washed 3 times with 500 μL of opti-MEM medium (Thermo Fisher Scientific, Waltham, MA, USA), and fixed with 200 μL of 4% PFA. Cell images from 5 different locations (the center and the 3, 6, 9, and 12 o’clock positions) of each well were captured by a camera attached to a fluorescence microscope. The average fluorescence intensity (arbitrary units) of these 5 images was calculated using the ImageJ, and the average intensity was used as a result for each well.

To confirm RPE-specific markers in Y-27632-treated iPS-RPE cells, the expression of VEGF and PEDF in iPS-RPE cells (TLHD2) was also evaluated by IHC. Y-27632-treated iPS-RPE or control RPE cells without Y-27632 were fixed with 4% PFA-PBS for 15 min at RT and permeabilized with 0.3% Triton X-100-PBS. Antihuman VEGF antibody (Abcam: catalog no. ab52917, Cambridge, UK), antihuman PEDF antibody (Abcam: catalog no. ab157207) or isotype control (rabbit) was used as the primary antibody, and Alexa Fluor-488 antirabbit IgG (Invitrogen: catalog no. A11034) was used as the secondary antibody. Cell nuclei were counterstained with DAPI. As in a previous report [30], we evaluated RPE staining by antibodies using confocal microscopy or by measurements of mean fluorescence intensity.

To detect phospho-myosin light chain 2 (pMLCII) in RPE cells, differentiated QHJIs01-01 iPSC-derived RPE cells were dissociated using TrypLE select (Thermo Fisher Scientific) and seeded on noncoated or Laminin E8 fragment (iMatrix511, Matrixosome)-coated 24-well plates (Thermo Fisher Scientific) with or without 10 µM Y-27632 in RPE stock solution. After a 15 min to 24 h incubation, RPE cells were fixed with 4% PFA at 4 °C for 30 min. These Y-27632-treated RPE cells were stained with Alexa Fluor 594-conjugated phalloidin (×1000: Thermo A12381) and pMLCII antibody (×200: Cell Signaling, # 3671S). Alexa Fluor 488-conjugated donkey antirabbit IgG was used as a secondary antibody for pMLCII staining. For the time-course assay, we prepared Y-27632-treated or nontreated RPE cells, and the incubation times were 15, 30, and 60 min and 3, 6, and 24 h. For the concentration assay, we also prepared Y-27632-treated RPE cells with 0.01, 0.1, 1, 10, or 100 µM of Y-27632. Cell nuclei were counterstained with DAPI. All samples were observed using a fluorescence microscope BZ-X800 (Keyence, Osaka, Japan).

For the confirmation of graft survival, after monkey sacrifice, the eyes that were collected at 8 weeks (DrpZ11) or at 6 months (Hekoyayu, Utsubo) were fixed and embedded in paraffin (Sigma-Aldrich-Aldrich). Enucleated eyes were fixed in formaldehyde (Super Fix, Kurabo, Osaka, Japan) and then embedded in paraffin (Sigma-Aldrich). We first performed hematoxylin and eosin (H&E) staining before IHC evaluation. The retinal sections were blocked with 5% goat serum. The sections with primary antibodies against immune cells/factors (CD3, CD4, CD8, CD20, CD40, MHC-class II, IgG, NKG2A, and Iba1) were incubated at 4 °C overnight. Staining of retinal sections was performed, and antibody information was previously reported [25,30]. In order to trace the grafts after transplantation, we stained RPE cells with PKH (Sigma-Aldrich-Aldrich). Images were acquired with a confocal microscope (LSM700, Zeiss, Oberkochen, Germany).

### 4.3. Quantitative RT-PCR and ELISA

To confirm RPE-specific markers or inflammatory cytokines/chemokines in Y-27632-treated iPS-RPE cells, we examined several factors by quantitative RT-PCR and ELISA. The mRNA expressions for *VEGF-A*, *PEDF*, *bestrophin-1 (Best1)*, *RPE65*, *Pax6*, *tyrosinase* (RPE-specific marker), *IL-6*, *TNF-α*, *CXCL11/I-TAC*, and *CCL2/MCP-1* (inflammatory cytokines/chemokines) were evaluated using quantitative RT-PCR. Total RNA was isolated from Y-27632-treated iPS-RPE cell lines (TLHD2). After cDNA synthesis, the expression of the above molecules and *β-actin* in triplicate samples was analyzed by qRT-PCR with a LightCycler 480 instrument by using qPCR Mastermix and highly specific Universal ProbeLibrary assays (all Roche Diagnostics, Mannheim, Germany). The tested primers and the Universal Probe for *Best1*, *RPE65*, *Pax6*, and *β-actin* (control), and the PCR conditions, are described in our previous report [27]. Other primers and probes are as follows: *VEGF-A*, L: cgcaagaaatcccggtataa, R: aaatgctttctccgctctga, probe #1; *PEDF*, L: gtgtggagctgcagcgtat, R: tccaatgcagaggagtagca, probe #57; *tyrosinase*, L: gctgccaatttcagctttaga, R: ccgctatcccagtaagtgga, probe #47; *IL-6*, L: gatgagtacaaaagtcctgatcca, R: ctgcagccactggttctgt, probe #40; *TNF-α*, L: cagcctcttctccttcctgat, R: gccagagggctgattagaga, probe #29; *CXCL11/I-TAC*, L: agtgtgaagggcatggcta, R: tcttttgaacatggggaagc, probe #76; *CCL2/MCP-1*, L: agtctctgccgcccttct, R: gtgactggggcattgattg, probe #40. Results indicate the relative expression of the molecules (ΔΔCt: control cells = 1). The concentrations of CCL2/MCP-1 in the supernatants of Y-27632-treated iPS-RPE cell lines (TLHD2) and control RPE cells without Y-27632 were measured by using human CCL2/MCP-1 ELISA (R&D Systems, Minneapolis, MN, USA).

### 4.4. Flow Cytometry

Proliferation of iPSC-derived RPE cells (Ff-I01, TLHD2, 453F2, 454E2: healthy donors) was evaluated by measuring Ki-67-positive cells. The medium (DMEM plus 10% FBS: total volume, 2 mL) was used. In the RPE cell-proliferation assay (Ki-67 FACS), 2 × 10^5^ iPSC-derived RPE cells were cultured for 3 days with or without 10 µM Y-27632 in 24-well culture plates without coating.

For FACS analysis of apoptosis, iPS-RPE cells (2 × 10^5^/well in 24-well plates; Ff-I01, TLHD2, and 253G1 lines) were treated with Y-27632 (10 µM) for 24 h. To evaluate the cell death, we measured annexin V-positive RPE cells. For staining, these RPE cells (Y-27632-treated or nontreated) were washed with cell-staining buffer (BioLegend, San Diego, CA, USA) and resuspended with annexin V binding buffer (BioLegend: catalog no. 420201). The cells were stained with FITC-labeled anti-annexin V antibody (BioLegend: catalog no. 640906) for 15 min at room temperature. Samples were analyzed using a FACSCanto^TM^ II flow cytometer (BD Biosciences). We performed similar experiments five times with different RPE cell lines, and the data were analyzed with FlowJo software (v9.3.1).

The expression of HLA-class II molecules on Y-27632-treated iPS-RPE cells (nontreated cells as a control) was evaluated. Before the assay, iPS-RPE cells were pretreated with recombinant human IFN-γ (100 ng/mL: R&D systems) for 48 h. In addition, before staining, these cells were incubated with a human Fc block (Miltenyi Biotec, Auburn, CA, USA) at 4 °C for 15 min. After human Fc block staining, these RPE cells were stained with FITC-labeled anti-HLA-class II antibodies (HLA-DR, DQ, DP: BioLegend: catalog no. 361705) at 4 °C for 30 min. RPE cells were also stained with FITC-labeled antimouse IgG at 4 °C for 30 min. RPE samples were analyzed using a FACSCanto^TM^ II.

To confirm the ability of phagocytosis in iPS-RPE cells, in Figure 4, iPS-RPE cells were cultured with FITC-labeled porcine shed photoreceptor rod outer segments (ROS, 10 µg/cm^2^) at 37 °C for 24 h. Control cells, which were iPS-RPE cells cultured without FITC-ROS, were also prepared [27].

### 4.5. Lymphocytes–Grafts Immune Reaction (LGIR) Test

PBMCs from several healthy donors (all HLA-mismatched against RPE cells) were purchased (stock PBMC: Precision for Medicine, Frederick, MD, USA) and prepared for the LGIR assay (Figure 3B). We also prepared two RPE cell groups, Y-27632-treated or nontreated TLHD2 iPS-RPE cells. The HLA haplotypes of the RPE lines were HLA-A*26:01/31:01, HLA-B*39:01/51:01, HLA-C*07:02/14:02, HLA-DRB1*09:01/15:01, HLA-DQB1*03:03/06:02, and HLA-DPB1*02:01/05:01. These PBMCs were cocultured with RPMI1640 medium containing 10% FBS, human recombinant IL-2 (BD), and other materials [26] in the presence of RPE cells in vitro. Details of the LGIR assay (Ki-67 FACS analysis), including antibody information, are described in a previous report [26,31]. As a positive control, EBV-transformed B cells were also prepared [26]. PBMC samples were analyzed using a FACSCanto^TM^ II.

### 4.6. Transplantation in In Vivo Monkey Models

Normal cynomolgus monkeys (names for monkey identification: Hekoayu, Utsubo, Ukigori) were purchased from Eve Bioscience Ltd. (Wakayama, Japan), and MHC control monkeys (haplotype identical: DrpZ11) were purchased from Ina Research Inc. (Nagano, Japan). For transplantation of monkey iPS-RPE cells (46a or 1121A1), a complete vitrectomy (Constellation^®^, Alcon, Geneva, Switzerland) was performed, and iPS-RPE cells (single-cell suspensions: 1–2 × 10^5^ cells/eye) were transplanted into the subretinal space as previously described in detail [25]. ROCK inhibitor Y-27632 (10 µM) was added to iPS-RPE cells, and Y-27632 graft RPE cells were inserted into the retina in the operation. As a control, RPE cells without Y-27632 were also prepared (Hekoayu right eye). After surgery, the operated monkeys’ heads were fixed for 3 h using anesthesia so that the recipients lied faced up for 3 h.

The transplanted cells were monitored by color fundus photographs, FA (both RetCamII^®^ and Clarity), and OCT (Nidek) at 1, 2, 4, 8 (DrpZ11 monkey), and 12 weeks and at 6 months (Hekoayu) after surgery. We transplanted allogeneic iPS-RPE cells into cynomolgus monkey eyes in an MHC haplotype-mismatched Hekoayu monkey with immunosuppressive medication with cyclosporin A. We monitored the concentration of cyclosporin A in the serum from the monkey before and after surgery. We also administrated a local steroid, subtenon triamcinolone acetonide injection (STTA), in the Utsubo monkey. We also checked MHC allele typing (Mafa-class I and class II) in the transplanted monkeys Hekoayu, Utsubo, Ukigori, and DrpZ11 [32]. We also measured the graft area (pixel/mm^2^) using NIH image J software after transplantation.

In order to test the toxicity of Y-27632 to retinal tissues, we injected 10 µM Y-27632 in a normal monkey eye (Ukigori left eye). As a control, we also injected only saline water without Y-27632 in the right eye. We then evaluated the toxicity by examination of color fundus, OCT, focal ERG (KOWA ER-80), and H&E staining after transplantation.

### 4.7. Statistical Evaluation

All statistical analyses were performed with Student’s *t* test (paired or unpaired, as appropriate). Values were considered statistically significant if *p* was less than 0.05.

## 5. Conclusions

The results of our study showed the effects of Y-27632 in iPSC-derived RPE cells. In vitro data showed suppression of apoptosis, promotion of cell adhesion, and higher proliferation and pigmentation of iPS-RPE cells. Moreover, Y-27632 increased the viability of the transplant without showing obvious retinal toxicity in iPS-RPE cell transplantation into monkey subretinal space in vivo. Thus, ROCK inhibitors may improve the engraftment of retinal-cell transplantation.

## Figures and Tables

**Figure 1 ijms-22-03237-f001:**
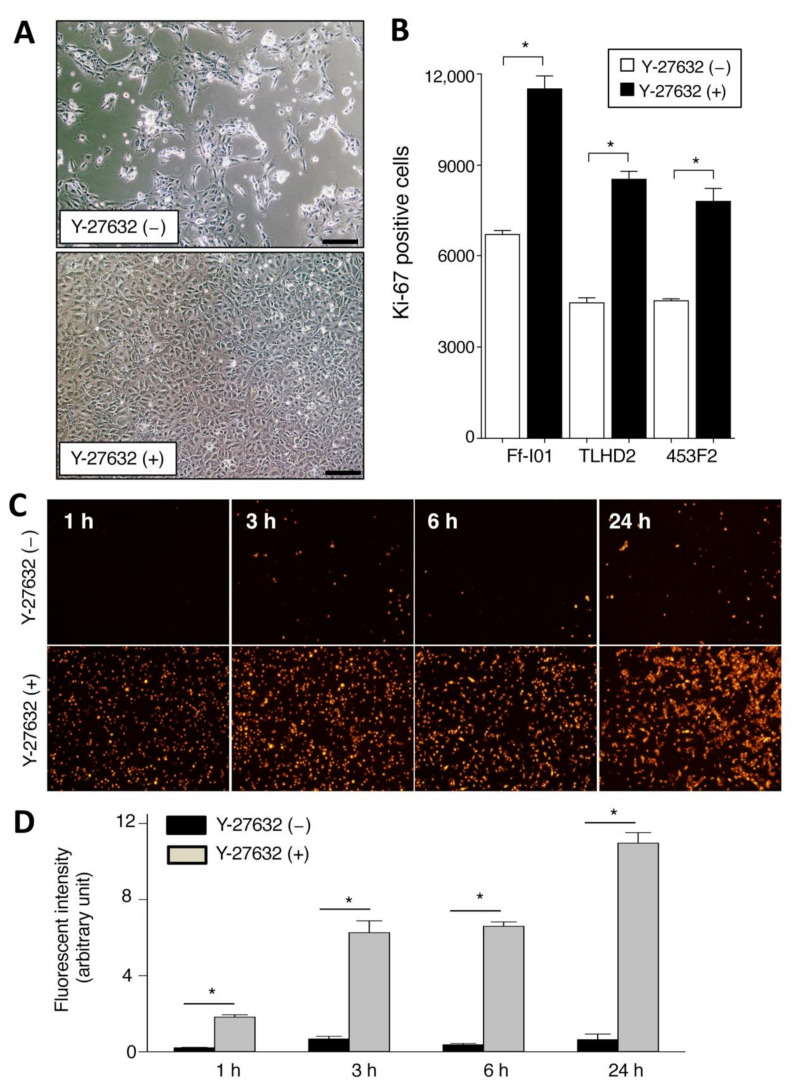
Cell proliferation and adhesion in established human iPS-cell-derived retinal pigment epithelium (RPE) cells that were cocultured with Y-27632. (**A**) RPE cells induced from iPS cells (TLHD2 line) clearly showed polygonal and mostly hexagonal proliferation for 3-day cultures with Y-27632 after reculturing (lower), whereas these iPS-RPE cells without Y-27632 showed poor proliferation (upper). Scale bar = 100 µm. (**B**) Proliferation of iPS-RPE cells such as lines Ff-I01, TLHD2, and 453F2 showed a similar pattern; proliferation of Y-27632-treated RPE cells was significantly higher than that of nontreated cells by Ki-67-positive FACS analysis. * *p* <0.05 between two groups. (**C**) Effects of Y-27632 to cell adhesion were examined. Representative cell images were presented. The iPS-RPE cells were labeled with PKH fluorescent dyes and seeded on the 24-well plates without coating materials (1.0 × 10^6^ cells/well) in the presence or absence of 10 μM Y-27632. These cells were incubated at 37 °C for indicated time periods, washed 3 times with 500 μL of medium, and fixed with 200 μL of 4% PFA. (**D**) Fluorescence intensity (arbitrary units) was calculated using the ImageJ (* *p* <0.05 between two groups., *n* = 2). Similar results were obtained from independent experiments on different days.

**Figure 2 ijms-22-03237-f002:**
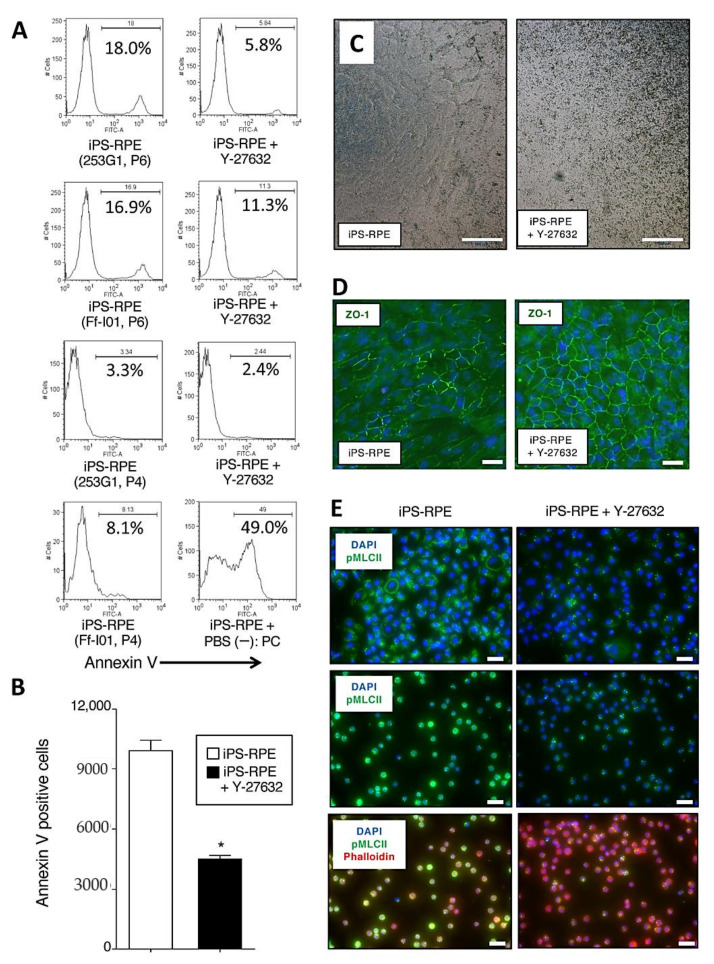
Apoptosis, pigmentation, and detection of ZO-1 and pMLCII in Y-27632-pretreated iPS-RPE cells. **(A**) iPS-RPE cells (253G1, Ff-I01 lines) were treated with Y-27632 (10 µM) for 24 h. Compared with nontreated RPE cells, Y-27632-treated RPE cells exhibited fewer annexin V-positive cells in flow cytometry. P4 or P6: passaged 4 or 6 times. As a positive control (PC), RPE cells in the presence of PBS (-) were also prepared. (**B**) There were significant differences between Y-27632-treated and nontreated iPS-RPE cells (253G1, P6) in annexin V FACS analysis. Data represent the mean ± SEM of three independent experiments, and we obtained similar results with other iPS-RPE cell lines (TLHD2 lines). * *p* <0.05 between two groups. (**C**) Pigmentation of the Y-27632-treated RPE cells (454E2 lines). Compared with nontreated cells, RPE cells in the presence of Y-27632 (added once per week in 4-week cultures) had much pigmentation. We did not obtain quantitative and statistics data, but we obtained similar results with other iPS-RPE cell lines (453F2, TLHD2 lines). These pictures are blight filed images by microscope. Scale bar = 500 µm. (**D**) Y-27632 treated iPS-RPE cells more clearly expressed ZO-1 (green) than nontreated cells. Scale bar = 50 µm. (**E**) Detection of phospho-myosin light chain 2 (pMLCII) in Y-27632-treated iPS-RPE cells. IHC for pMLCII (green) in human iPS-RPE cells without (left) or with (right) 10 µM Y-27632. Cells were counterstained with phalloidin (red) and DAPI (blue). These RPE cells were seeded onto Laminin E8 fragment (iMatrix511: higher panels)-coated and noncoated plates (middle panels). The lower panels also show staining with phalloidin (red) together with pMLCII (green). We obtained similar results with other iPS-RPE cell lines (453F2, TLHD2 lines). Scale bar = 50 µm.

**Figure 3 ijms-22-03237-f003:**
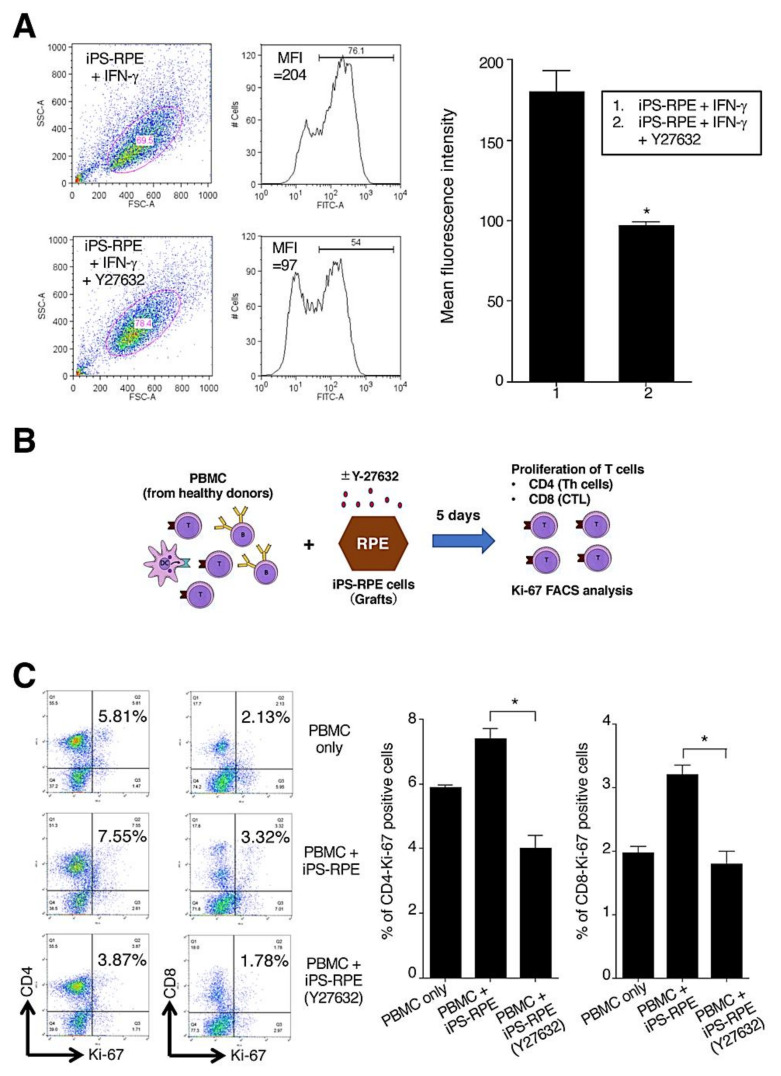
Y-27632-treated iPS-RPE cells have low immunogenicity. (**A**) Expression of HLA-class II molecules on Y-27632-treated iPS-RPE cells (TLHD2 lines). Before the assay, RPE cells were pretreated with recombinant IFN-γ (100 ng/mL) for 48 h. Compared with nontreated cells, the expression of HLA-class II (DR, DQ, DP) on Y-27632-treated iPS-RPE cells was poor according to FACS. The right panel shows that there were significant differences between Y-27632-treated vs. nontreated RPE cells in mean fluorescence intensity (MFI) of the expression of HLA-class II. Data represent the mean ± SEM of three independent experiments. We obtained similar results with other iPS-RPE cell lines (453F2, 454E2 lines). (**B**) Schema of LGIR assay are shown: PBMC from healthy donors were cocultured with graft iPS-RPE cells. Before the assay, RPE cells were pre-cultured with (or without) 10 µM Y-27632. After coculturing for 5 days, PBMC were harvested for Ki-67 FACS analysis (e.g., proliferation of CD4^+^ and CD8^+^ T cells). Th, T helper cells; CTL, cytotoxic T cells. (**C**) Results of the LGIR test for allogeneic T lymphocytes and Y-27632-treated iPS-RPE cells (TLHD2 lines). PBMCs including T lymphocytes were prepared from healthy donors (all HLA mismatched against RPE cells). For nontreated RPE cells in the LGIR test, there was great proliferation of CD4^+^/Ki-67^+^ and CD8^+^/Ki-67^+^ in PBMC as compared with PBMC only. However, there was poor proliferation of proliferative T cells (CD4 and CD8) against Y-27632-treated RPE cells. The right graph shows %CD4 or %CD8-Ki-67-positive cells (i.e., proliferation of T cells) in FACS analysis (*n* = 3, three independent experiments). The proliferative response of CD4^+^ helper T cells or CD8^+^ cytotoxic T cells to Y-27632-treated RPE cells was significantly reduced compared with Y-27632 nontreated RPE cells. * *p* < 0.05 between two groups.

**Figure 4 ijms-22-03237-f004:**
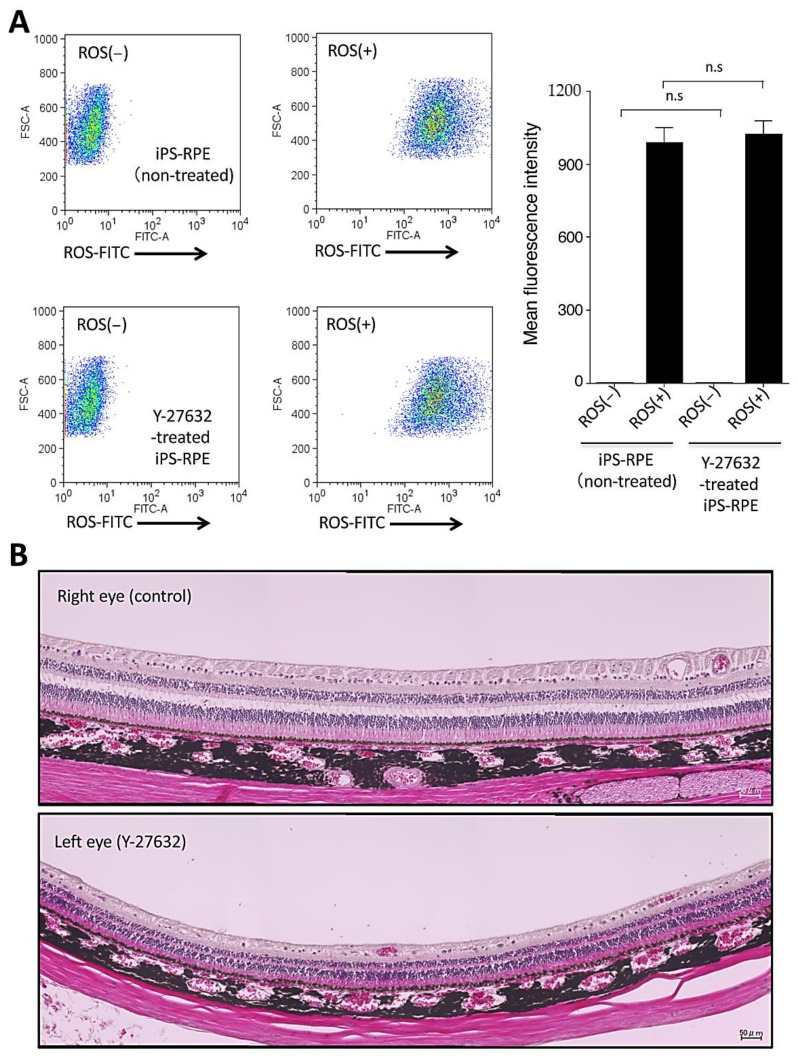
Toxicity of iPS-RPE cells and retinal cells by Y-27632 treatment. (**A**) Measurement of phagocytosis by Y-27632-treated iPS-RPE cells (253G1 lines). iPS-RPE cells were cultured with FITC-labeled porcine shed photoreceptor rod outer segments (ROS) at 37 °C for 24 h and analyzed using flow cytometry. Control cells, which were iPS-RPE cells cultured without FITC-ROS, were used to obtain baseline fluorescence (left graphs), and iPS-RPE cells without ROS at 37 °C were also analyzed. The right panel shows that there were no significant differences between Y-27632-treated vs. nontreated RPE cells in the expression of FITC-ROS. Data represent the mean ± SEM of three independent experiments, and we obtained similar results with other iPS-RPE cell lines (TLHD2 lines). n.s., not significant. (**B**) Toxicity of in vivo retinas in a cynomolgus monkey by Y-27632 treatment. We administrated Y-27632 in saline into the left eye, and saline only as a control into the right eye, and evaluated the retinas including the RPE layer using H&E staining of retinal sections. Similar to control retina (**upper**), retinal sections with Y-27632 administration showed no damage with H&E staining (**lower**).

**Figure 5 ijms-22-03237-f005:**
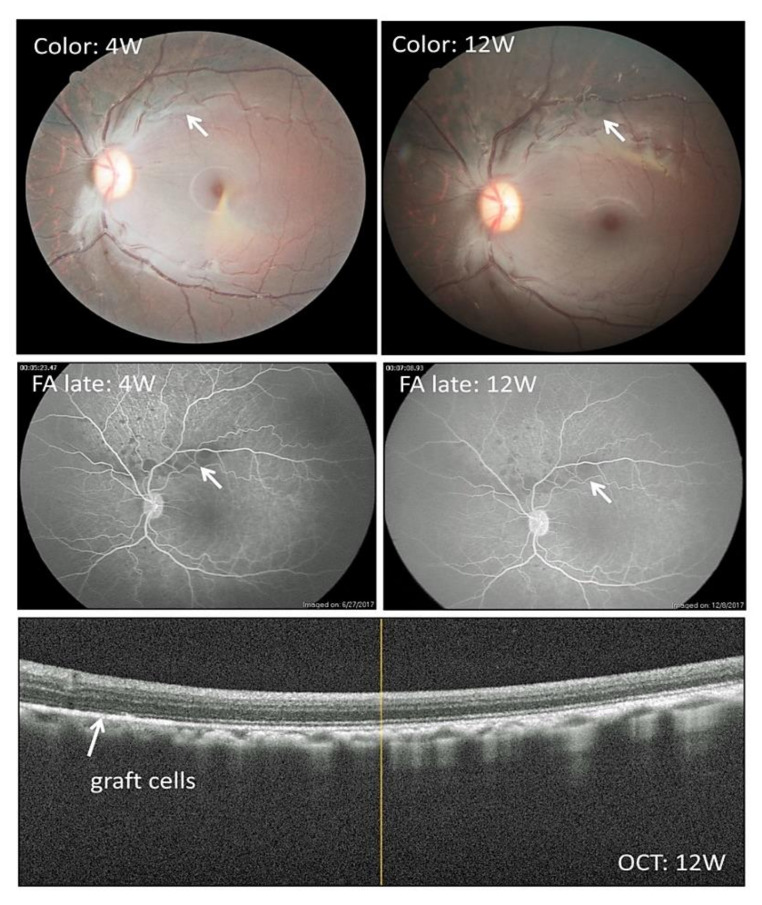
Clinical features after iPS-RPE cell transplantation together with Y-27632 in an allogeneic monkey. Representative color fundus, fluorescence angiography (FA), and OCT images are shown for the allogeneic transplantation using a normal healthy monkey (Hekoayu) and a monkey with iPS-RPE cells (single cell suspensions; 46a line). Before transplantation, we administrated cyclosporine A. In the operation, we injected iPS-RPE cells together with Y-27632 in the subretinal space. Pigmented cells became visible by 4 weeks (4W) after the transplantation, with a stable pigmented area observed at 12 weeks (12W) in this case (white arrows). There were no rejection signs in FA (e.g., no leakages from the graft cells: middle panels). In OCT (lower panel), we can see graft RPE cells in the subretinal space (white arrow).

**Figure 6 ijms-22-03237-f006:**
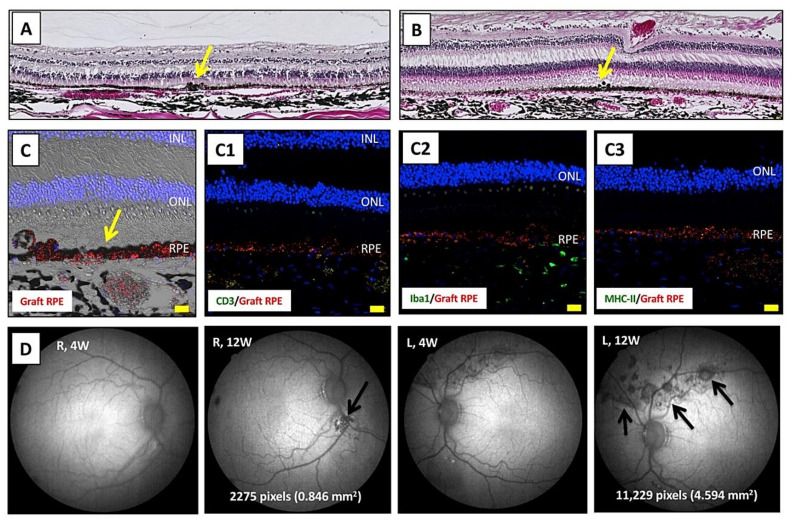
Graft survival, expanded graft cells, and no inflammation in Y-27632 RPE cell-transplanted retinas. (**A**) H&E staining shows RPE cell graft survival in the right eye, but grafted cells look like clumps (yellow arrow). (**B**) In contrast, in the left eye (Y-27632 RPE cell transplantation), there was RPE cell graft survival and expanded iPS-RPE cells in the subretinal space. The expanded RPE cells became sheetlike (yellow arrow). (**C**) Confocal microscopy of IHC shows PKH-labeled RPE cells (graft RPE, red) through the retinal sections of the left eye. Unlike an immune-reactive retina, there were no CD3^+^ T cells (**C1**, green), few Iba1^+^ microglia/macrophages (**C2**, green), and no MHC-II^+^ antigen-presenting cells (**C3**, green) invading the retina and choroid throughout the sections, and there was graft survival of transplanted RPE cells (red). INL, inner nuclear layer; ONL, outer nuclear layer; RPE, retinal pigment epithelium. Scale bar = 20 µm. (**D**) AF images show a dark area that corresponds to the grafted RPE cells after transplantation (arrows). Compared with transplantation without Y-27632 (right eye), the cell graft area was 5-fold greater with Y-27632 (left eye).

**Figure 7 ijms-22-03237-f007:**
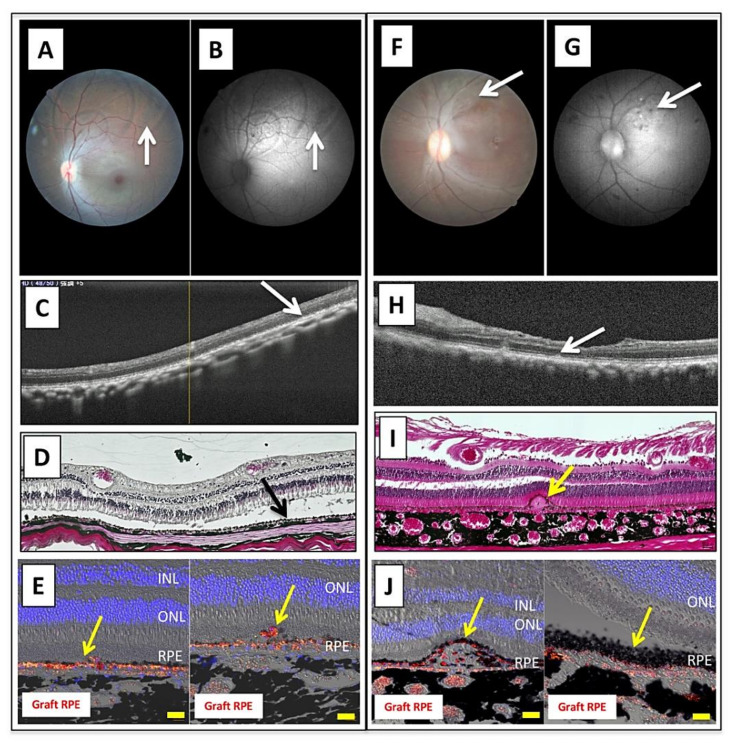
Transplantation of allogeneic iPS-RPE cells with Y-27632 in MHC haplotype-matched or steroid-administrated monkeys. (**A**–**E**) MHC haplotype-matched donor: (**A**) color fundus, (**B**) AF fundus, (**C**) OCT, (**D**) H&E staining, (**E**) IHC staining. Pigmented graft RPE cells are seen in the left eye (arrows). Grafted cells survived and there were no rejection signs in OCT (**C**), H&E (**D**), and IHC staining (**E**). In addition, we found PKH-labeled iPS-RPE cells (graft RPE, red) through the retinal sections. Scale bar = 20 µm. (**F**–**J**) MHC-mismatched steroid administration donor: (**F**) color fundus, (**G**) AF fundus, (**H**) OCT, (**I**) H&E staining, (**J**) IHC staining. We obtained similar results for the MHC-mismatched donor with immunosuppressive drugs and local steroid administration. In these examinations such as color fundus (**F**), AF (**G**), OCT (**H**), H&E staining (**I**), and IHC staining (**J**), there was graft survival of iPS-RPE cells in the subretinal space. We also found PKH-labeled iPS-RPE cells (graft RPE, red) throughout the retinal sections. Scale bar = 20 µm.

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
