# Peer review of "A ROCK Inhibitor Promotes Graft Survival during Transplantation of iPS-Cell-Derived Retinal Cells"

_ijms, 2021, doi:10.3390/ijms22063237_

Round 1

Reviewer 1 Report

The purpose of this study is to demonstrate that  the ROCK inhibitor Y-27632 improves the development of iPSC-derived RPE cells and their survival after transplantation. Y-327632 significantly enhanced cell proliferation and cell adhesion, whereas cell apoptosis was reduced. Y-327632 also reduced the expression of inflammatory cytokines, expression of MHC-II, and suppressed the activation of lymphocytes in vitro. Transplantation of allogenic iPS-RPE cell suspensions (MHC-II mismatched) together with Y-327632 resulted in transplants covering a larger area than transplants of iPSC-RPE without Y-327632. A cell tracker (PKH) was used to identify the transplanted RPE cells. Y-327632-treated RPE cells appeared to form “sheets” in the subretinal space. Injection of Y-327632 (without cells) into the subretinal space did not cause any damage to the tissues and did not cause any change to ERGs.  

General comments:

This is a well-written paper, covering different aspects of Y-327632 effects on iPS-RPE. The reviewer has only minor comments.

Y-27632 is a well-established substance to increase cell survival. It has also shown a beneficial effect on RPE cell culture (including PSC-derived RPE) in many different studies.

The authors are citing few of the many studies that have shown a beneficial effect of Y-27632 treatment on RPE cells (other than iPSC-derived). This appears to be an extension of a study published by some of the same authors in 2020 (Ye et al. 2020, Scientific Reports) which was however not cited.

Specific comments:

  1. 1, line 42: typo “cornel”
  2. 1, line 45-46: “there are no studies so far regarding the usage of Y-327632 for iPS/ES-RPE cell transplantation.” – This is not entirely correct. Sugita et al 2016 used this inhibitor, but did not go into the mechanism of action. Zheng et al. (2004) studied the effect of Y-327632 on transplantation of primary RPE cells.
  3. 4, line 96: MLCII (phospho-myosin light chain 2) should be explained the first time it is mentioned.
  4. 14, line 365: “Since RPE cells exhibit immunogenic” – sentence is incomplete. Proposal: “Since RPE cells exhibit immunogenic properties …”.

Figure S4D: the cone inner segments appear to be reduced in the Y-327632 section. The authors state that there is no difference.

Author Response

General comments:

This is a well-written paper, covering different aspects of Y-327632 effects on iPS-RPE. The reviewer has only minor comments. Y-27632 is a well-established substance to increase cell survival. It has also shown a beneficial effect on RPE cell culture (including PSC-derived RPE) in many different studies. The authors are citing few of the many studies that have shown a beneficial effect of Y-27632 treatment on RPE cells (other than iPSC-derived). This appears to be an extension of a study published by some of the same authors in 2020 (Ye et al. 2020, Scientific Reports) which was however not cited.

Response: We have added the sentences with the reference as following: To promote RPE differentiation from human iPSC, Ye et al used and added 10 μM Y-27632 to culture medium for 18 days (doi: 10.1038/s41598-020-70979-y)(revised manuscript, lines 48-50).

Specific comments:

  1. 1, line 42: typo “cornel”

Response: We have revised it (revised manuscript, line 44).

  1. 1, line 45-46: “there are no studies so far regarding the usage of Y-327632 for iPS/ES-RPE cell transplantation.” – This is not entirely correct. Sugita et al 2016 used this inhibitor, but did not go into the mechanism of action. Zheng et al. (2004) studied the effect of Y-327632 on transplantation of primary RPE cells.

Response: We have added the sentence with the reference (lines 47-48).

  1. 4, line 96: MLCII (phospho-myosin light chain 2) should be explained the first time it is mentioned.

Response: As per your suggestion, we added it (line 94).

  1. 14, line 365: “Since RPE cells exhibit immunogenic” – sentence is incomplete. Proposal: “Since RPE cells exhibit immunogenic properties …”.

Response: We changed the sentence (lines 376-377).

Figure S4D: the cone inner segments appear to be reduced in the Y-27632 section. The authors state that there is no difference.

Response: We think that there was difference which section we prepared. So, we changed from original one to other one (the Y-27632 section). Please see new Figure S4-D. Thank you so much for your helpful comments.

Reviewer 2 Report

Dear authors,

The paper has a high potential and it is really interesting your comments about the field.

My minor comment is about in the Figure 3, the author should add a schematic representation to understand better the interesting experiment

Author Response

Dear authors,

The paper has a high potential and it is really interesting your comments about the field.

My minor comment is about in the Figure 3, the author should add a schematic representation to understand better the interesting experiment.

Response: As per your suggestion, we have added schematic representation to understand better the experiment (LGIR assay). Please see new Figure 3B. Thank you for your helpful comments.

Reviewer 3 Report

In this manuscript “ A ROCK inhibitor promotes graft survival during transplantation of iPS cell-derived retinal cells ” by Masaaki Ishida et al., the authors combine both in vitro and in vivo models to interrogate the impact of the ROCK inhibitor Y- 27632 in promoting graft survival during transplantation of iPS-RPE cells. The authors showed that the Y- 27632 ROCK inhibitor improved iPS-RPE cells properties, including viability, proliferation, morphology and expression markers and moreover, Y- 27632 in vivo administration increased the viability of the iPS-RPE cells transplanted without retinal toxicity in monkey subretinal space in vivo.

The study is generally well design, the statistical analysis and methodologies are adequate. The results obtained are novel and conclusions are well supported by the presented data. However, the entire manuscript should be revised as it contains some unclear sentences and incorrections and some aspects of the Figures and manuscript structure should be improved (as described below). Moreover, the Experimental Procedures section should be completely reformulated as to improve the clarity for the readers.

Overall, I consider that this manuscript provides a major advance in the field of retinal transplantation by combining the ROCLK inhibitor with iPS-RPE cells suspensions. Therefore, I consider that, after providing the minor revisions and corrections depicted bellow, this manuscript overall is of high interest for the readers and therefore, it merits publication

A list of specific concerns and minor revisions is provided bellow.

Specific concerns:

Line 19 – in the abstract, need to contextualize the rational for using ROCK inhibitors

Line 48 – in formulation of objective of the study, there is no clear indication of the in vivo model used (“on transplantation of iPS-RPE cell suspensions into the subretinal space.”).

Line 59 the subsection title (“2.1. Cell proliferation in human iPS-RPE cells with Y-27632”) only refers to part of the presented results.

Line 62 – clarify the use of polygonal and hexagonal as description for proliferation (“presence of Y-27632 clearly showed polygonal and hexagonal proliferation, whereas cells”), in the legend the authors used the term morphology instead (“hexagonal morphology”).

Regarding Figure 1D, not clear if the fluorescent results arise from flow cytometry analysis as stated in the text (“PKH-positive cell determination in FACS analysis 69 (Figure 1D)” – line 69 or from the quantification of the images from Figure 1C as indicated in the Figure legend (“D) Fluorescence intensity (arbitrary units) was calculated using the ImageJ “) as Image J is not a commonly used software for the quantification of flow cytometry data and rather used for pixel quantification of fluorescent images.

Line 92 – “contained a lot of melanin” – reformulate

Line 84 – In the “2.2. Inhibition of apoptosis and the promotion of pigmentation in Y-27632-treated iPS-RPE cells” section, there is no allusion or description of the results from panel Figure 2E.

In the Figure 2, there are missing labels in the images; 2C is missing the identification of the staining, and in 2E there is no identification of cells and treatment.

Supplementary Figure S2 – statistical analysis in graphs S2A, S2B and S2E is missing. The symbol for micromolar (m) labeling drug concentrations in the graph’s legends, is not correctly displayed.

Line 147 – “there was significant proliferation of CD8+/Ki-67+ (proliferative cytotoxic T cells) in peripheral blood mononuclear cells 148 (PBMC) with Y-27632-treated RPE cells compared with the PBMC plus RPE cells (no Y- 27632).” – this statement should be clarified as it seems to indicate that there is more proliferative cytotoxic T cells in the presence of Y-27632-treated RPE cells.

Line 168 –sub-section 2.5 is not necessary and description of this results should be at the end of sub-section 2.4 as it alludes to the direct impact of Y-27632 in suppressing proliferation and activation of lymphocytes in vitro.

Line 176 – The title of the subsection “2.6. Toxicity of iPS-RPE cells and retinal cells by Y-27632 treatment.” should also refer to the use of an animal model for in vivo evaluation of toxicity in the retinas.

Line 191 – In Supplementary Figure S3, the PEDF immunofluorescence images used for Figure S3D quantification are lacking. Since the graphs display an impact of Y- 27632 in the expression of PEDF, it will be important to show the original images.

Line 224 – “2.7. Transplantation of allogeneic iPS-RPE cells together with Y-27632 in an in vivo animal 224 model “ – please reformulate to specify which animal model was used.

Line 237 – at 4 and 24 weeks (Figure S5).” - Should indicate the Figure panel (A, B,..).

Line 251 – “2.8. Graft survival and expanded graft RPE cells in retinas with Y-27632 RPE cell 251 transplantation. “ – please reformulate.

Line 253 - Authors may consider fusing the sub-sections 2.7 and 2.8, as section 2.8 results description is a clear continuation of sub-section 2.7 “After sacrifice, we performed H&E staining using retinal sections from transplanted “.

For consistency, in Figure 6, label identification of cells and treatment in panels and B staining used (PKH- 275 labeled RPE cells (red) in panel C. The same for Figure 7E, J.

In Figure 6C, as the last 3 images are stained of different proteins (CD3, Iba1, MHCII), consider using a new label for panel identification (as C’-C’’’).

Line 258 – “the in vivo expanded RPE 258 cells were seen through the retinal sections from the left eyes (Figure S5-D).” – clarify if those cells are Y-27632 RPE cell transplanted.

 Line 267 – “Compared with transplantations without Y- 267 27632, the cell graft area was 5-fold greater in transplantations with Y-27632” – if possible provide the quantification data supporting this statement as Supplementary.

Line 330 – “These results suggest that Y-27632 may provide for faster 330 and more efficient protocols “ – please reformulate

Line 365 – “Since RPE cells exhibit immunogenic [27]” – please clarify (immunogenicity, elicit and immunogenic response).

Line 370 – The 4. Experimental Procedures section should be completely reformulated.

The different sub-sections should refer to procedures of different techniques and assays used (Cell Culture and Reagents, Immunohistochemistry and Microscopy, Quantitative Real-Time Polymerase Chain Reaction (RT-qPCR), Flow cytometry) instead of being divided by the Manuscript Figures.

Specifically, try to aggregate all the procedure of in vitro iPS-derived RPE cell culture establishment, maintenance and drug treatment in the sub-section 4.1. (Preparation of cultures of iPS-derived RPE (iPS-RPE) cells), and maintain minimal detail for this procedure in the subsequent sub-sections.

4.2. Morphology, proliferation, and pigmentation of iPS-RPE cells in the presence of Y-27632 – should be eliminated and procedures included as part of sub-section 4.1., IHC and microscopy and flow cytometry analysis sub-sections.

4.3. Cell-to-cell adhesion assays for Y-27632-treated iPS-RPE cells should be included as part of sub-section 4.1 and IHC and microscopy sub-sections.

4.4. Apoptosis assays for Y-27632-treated iPS-RPE cells” – should be included in the flow cytometry sub-section.

4.5. pMLCII staining for Y-27632-treated iPS-RPE cells procedures should be included as part of IHC and microscopy procedures.

4.6. Expression of HLA-class II molecules on Y-27632-treated iPS-RPE cells procedures should be included as part of sub-section 4.1. (iPS-RPE cells were pretreatment) and flow cytometry sub-section.

4.7. Lymphocyte-Graft cells Immune Reaction (LGIR) test – should contain procedures for PBMCs culture; and FACS analysis should go to flow cytometry sub-section.

4.8. Phagocytosis by Y-27632-treated iPS-RPE cells – procedures should be allocated to cell culture (4-1 sub-section) and flow cytometry sub-sections.

4.9. Expression of RPE-specific markers and inflammatory cytokines in Y-27632-treated iPS-RPE cells. - procedures should be allocated to RT-PCR, ELISA and IHC sub-sections.

2.10. Transplantation in an in vivo animal model; 2.11. Immunohistochemistry (IHC) and 2.12. Statistical evaluation - should be numbered with 4.

Line 509 -  2.11. Immunohistochemistry (IHC) – should contain additionally all IHC procedures from in vitro iPS-RPE cells, together with procedures for microscopy acquisition and analyses.

In Conclusion, Line 528 – “In fact, in our past clinical study with al- 528 logeneic HLA-matched iPS-RPE cell transplantation [26, 30], there was the problem in that 529 iPS-RPE cell suspensions regurgitated into the vitreous cavity and thus formed an epiretinal membrane. Our current results may play a part in solving this issue”.- please clarify this sentence.

Minor revisions:

Line 129 – “had reduced mRNA expressions” – expression

Line 242 – The legend of Figure 5 is edited as main text.

Line 323 – “morphology and expressions” – expression

Line 378 – “The ROCK inhibitor Y-27632 (1, 10, 50, or 100 μM: Wako) were added” – was added

Author Response

Specific concerns:

Line 19 – in the abstract, need to contextualize the rational for using ROCK inhibitors

Response: We have revised it in abstract (revised manuscript, lines 19-21).

Line 48 – in formulation of objective of the study, there is no clear indication of the in vivo model used (“on transplantation of iPS-RPE cell suspensions into the subretinal space.”).

Response: As per your concerns, we have added the sentences for the objective of the study (lines 54-56).

Line 59 the subsection title (“2.1. Cell proliferation in human iPS-RPE cells with Y-27632”) only refers to part of the presented results.

Response: As your concerns, we have revised the subsection title (line 68).

Line 62 – clarify the use of polygonal and hexagonal as description for proliferation (“presence of Y-27632 clearly showed polygonal and hexagonal proliferation, whereas cells”), in the legend the authors used the term morphology instead (“hexagonal morphology”).

Response: We are sorry for the confusing. To clarify, we have omitted “morphology” and revised some sentences in the text and figure legend.

Regarding Figure 1D, not clear if the fluorescent results arise from flow cytometry analysis as stated in the text (“PKH-positive cell determination in FACS analysis 69 (Figure 1D)” – line 69 or from the quantification of the images from Figure 1C as indicated in the Figure legend (“D) Fluorescence intensity (arbitrary units) was calculated using the ImageJ “) as Image J is not a commonly used software for the quantification of flow cytometry data and rather used for pixel quantification of fluorescent images.

Response: We are so sorry for the confusing again. That was just error. We have omitted “FACS analysis” to the explanation of Figure 1C, D.

Line 92 – “contained a lot of melanin” – reformulate

Response: We have omitted “contained a lot of melanin” and revised it (Line 102).

Line 84 – In the “2.2. Inhibition of apoptosis and the promotion of pigmentation in Y-27632-treated iPS-RPE cells” section, there is no allusion or description of the results from panel Figure 2E.

Response: We have revised the subsection title (lines 93-94).

In the Figure 2, there are missing labels in the images; 2C is missing the identification of the staining, and in 2E there is no identification of cells and treatment.

Response: Figure 2C is no staining data. But, we have added the explanation in Figure 2C legend. In addition, we have revised Figure 2E as per your suggestion.

Supplementary Figure S2 – statistical analysis in graphs S2A, S2B and S2E is missing. The symbol for micromolar (m) labeling drug concentrations in the graph’s legends, is not correctly displayed.

Response: We have revised the labeling drug concentrations in the graph’s legends, and we have added the data for statistical analysis in Supplementary Figure S2-A, B and E. Please see new Figure S2.

Line 147 – “there was significant proliferation of CD8+/Ki-67+ (proliferative cytotoxic T cells) in peripheral blood mononuclear cells 148 (PBMC) with Y-27632-treated RPE cells compared with the PBMC plus RPE cells (no Y- 27632).” – this statement should be clarified as it seems to indicate that there is more proliferative cytotoxic T cells in the presence of Y-27632-treated RPE cells.

Response: We are sorry for the confusing. These are just error sentences. Therefore, we have revised these sentences in new Fig. 3C (Lines 151-159, lines 167-170).

Line 168 –sub-section 2.5 is not necessary and description of this results should be at the end of sub-section 2.4 as it alludes to the direct impact of Y-27632 in suppressing proliferation and activation of lymphocytes in vitro.

Response: We agree with your comments. So, we have revised it.

Line 176 – The title of the subsection “2.6. Toxicity of iPS-RPE cells and retinal cells by Y-27632 treatment.” should also refer to the use of an animal model for in vivo evaluation of toxicity in the retinas.

Response: We have revised the subsection title, new 2.5. (lines 187-188).

Line 191 – In Supplementary Figure S3, the PEDF immunofluorescence images used for Figure S3D quantification are lacking. Since the graphs display an impact of Y- 27632 in the expression of PEDF, it will be important to show the original images.

Response: As per your suggestions, we have added the data for PEDF IHC (original images) of Y-27632-treated iPS-RPE cells (new Fig. S3-C & lines 200-205).

Line 224 – “2.7. Transplantation of allogeneic iPS-RPE cells together with Y-27632 in an in vivo animal 224 model “ – please reformulate to specify which animal model was used.

Response: As per your suggestions, we have reformulated the sentence (2.6., lines 236-237).

Line 237 – “at 4 and 24 weeks (Figure S5).” - Should indicate the Figure panel (A, B,..).

Response: In the text, we have indicated the panels of Figure S5 (Lines 247-253).

Line 251 – “2.8. Graft survival and expanded graft RPE cells in retinas with Y-27632 RPE cell 251 transplantation. “ – please reformulate.

Response: As per your suggestions, we have reformulated the sentence (sub-sections 2.7., lines 292-293).

Line 253 - Authors may consider fusing the sub-sections 2.7 and 2.8, as section 2.8 results description is a clear continuation of sub-section 2.7 “After sacrifice, we performed H&E staining using retinal sections from transplanted “.

Response: As per your suggestions, we are fusing sub-sections 2.7 and 2.8, and that is now new sub-sections 2.7. Then, we have changed the title of sub-section (Lines 292-293). Moreover, we added the sentence “After sacrifice, we performed H&E staining using retinal sections from transplanted“ in the text  (Lines 299-300).

For consistency, in Figure 6, label identification of cells and treatment in panels and B staining used (PKH- 275 labeled RPE cells (red) in panel C. The same for Figure 7E, J.

Response: As per your suggestions, we have changed some panels and legends of Figure 6 & 7.

In Figure 6C, as the last 3 images are stained of different proteins (CD3, Iba1, MHCII), consider using a new label for panel identification (as C’-C’’’).

Response: As per your suggestions, we have used new labels for panel identification as Figure 6C’, C’’, and C’’’.

Line 258 – “the in vivo expanded RPE 258 cells were seen through the retinal sections from the left eyes (Figure S5-D).” – clarify if those cells are Y-27632 RPE cell transplanted.

Response: To clarify, we have added it (Lines 267-268).

Line 267 – “Compared with transplantations without Y- 267 27632, the cell graft area was 5-fold greater in transplantations with Y-27632” – if possible provide the quantification data supporting this statement as Supplementary.

Response: To clarify, we have added the explanation in the text (Lines 277-279).

Line 330 – “These results suggest that Y-27632 may provide for faster 330 and more efficient protocols “ – please reformulate

Response: We have revised the sentence (Lines 341-342).

Line 365 – “Since RPE cells exhibit immunogenic [27]” – please clarify (immunogenicity, elicit and immunogenic response).

Response: Another reviewer (reviewer #1) raised similar concerns. So, we had revised to “immunogenic properties” (Lines 377-378).

Line 370 – The 4. Experimental Procedures section should be completely reformulated.

The different sub-sections should refer to procedures of different techniques and assays used (Cell Culture and Reagents, Immunohistochemistry and Microscopy, Quantitative Real-Time Polymerase Chain Reaction (RT-qPCR), Flow cytometry) instead of being divided by the Manuscript Figures.

Specifically, try to aggregate all the procedure of in vitro iPS-derived RPE cell culture establishment, maintenance and drug treatment in the sub-section 4.1. (Preparation of cultures of iPS-derived RPE (iPS-RPE) cells), and maintain minimal detail for this procedure in the subsequent sub-sections.

Response: The Method part has been significantly restructured as per your suggestions.

4.2. Morphology, proliferation, and pigmentation of iPS-RPE cells in the presence of Y-27632 – should be eliminated and procedures included as part of sub-section 4.1., IHC and microscopy and flow cytometry analysis sub-sections.

4.3. Cell-to-cell adhesion assays for Y-27632-treated iPS-RPE cells should be included as part of sub-section 4.1 and IHC and microscopy sub-sections.

4.4. Apoptosis assays for Y-27632-treated iPS-RPE cells” – should be included in the flow cytometry sub-section.

4.5. pMLCII staining for Y-27632-treated iPS-RPE cells procedures should be included as part of IHC and microscopy procedures.

4.6. Expression of HLA-class II molecules on Y-27632-treated iPS-RPE cells procedures should be included as part of sub-section 4.1. (iPS-RPE cells were pretreatment) and flow cytometry sub-section.

4.7. Lymphocyte-Graft cells Immune Reaction (LGIR) test – should contain procedures for PBMCs culture; and FACS analysis should go to flow cytometry sub-section.

4.8. Phagocytosis by Y-27632-treated iPS-RPE cells – procedures should be allocated to cell culture (4-1 sub-section) and flow cytometry sub-sections.

4.9. Expression of RPE-specific markers and inflammatory cytokines in Y-27632-treated iPS-RPE cells. - procedures should be allocated to RT-PCR, ELISA and IHC sub-sections.

2.10. Transplantation in an in vivo animal model; 2.11. Immunohistochemistry (IHC) and 2.12. Statistical evaluation - should be numbered with 4.

Line 509 -  2.11. Immunohistochemistry (IHC) – should contain additionally all IHC procedures from in vitro iPS-RPE cells, together with procedures for microscopy acquisition and analyses.

Response: The Method part has been significantly restructured and we have revised them (4. Experimental procedures section).

In Conclusion, Line 528 – “In fact, in our past clinical study with al- 528 logeneic HLA-matched iPS-RPE cell transplantation [26, 30], there was the problem in that 529 iPS-RPE cell suspensions regurgitated into the vitreous cavity and thus formed an epiretinal membrane. Our current results may play a part in solving this issue”.- please clarify this sentence.

Response: To clarify that, we have omitted these sentences and revised the conclusions (Lines 544-550).

Minor revisions:

Line 129 – “had reduced mRNA expressions” – expression

Line 242 – The legend of Figure 5 is edited as main text.

Line 323 – “morphology and expressions” – expression

Line 378 – “The ROCK inhibitor Y-27632 (1, 10, 50, or 100 μM: Wako) were added” – was added

Response: We have revised all of them as per your suggestions. Thank you for your helpful comments.